**Data Availability Statement:** All relevant data are within the manuscript and its Supporting information files.

# Obesity defined by body mass index and waist circumference and risk of total knee arthroplasty for osteoarthritis: A prospective cohort study

Yuan Z. Lim[1]☯, Yuanyuan Wang[1]☯, Flavia M. Cicuttini[1], Graham G. Giles[1,2,3], Stephen Graves[4], Anita E. Wluka🆔[1], Sultana Monira Hussain🆔[1]*

1 Department of Epidemiology and Preventive Medicine, School of Public Health and Preventive Medicine, Monash University, Melbourne, Victoria, Australia, 2 Cancer Epidemiology Division, Cancer Council Victoria, Melbourne, Victoria, Australia, 3 Centre for Epidemiology and Biostatistics, School of Population and Global Health, The University of Melbourne, Melbourne, Victoria, Australia, 4 Australian Orthopaedic Association National Joint Replacement Registry, Discipline of Public Health, School of Population Health & Clinical Practice, University of Adelaide, Adelaide, South Australia, Australia

☯ These authors contributed equally to this work.
* monira.hussain@monash.edu

## Abstract

### Objective

To examine the risk of total knee arthroplasty (TKA) due to osteoarthritis associated with obesity defined by body mass index (BMI) or waist circumference (WC) and whether there is discordance between these measures in assessing this risk.

### Methods

36,784 participants from the Melbourne Collaborative Cohort Study with BMI and WC measured at 1990–1994 were included. Obesity was defined by BMI ($\geq$30 kg/m²) or WC (men $\geq$102cm, women $\geq$88cm). The incidence of TKA between January 2001 and December 2018 was determined by linking participant records to the National Joint Replacement Registry.

### Results

Over 15.4±4.8 years, 2,683 participants underwent TKA. There were 20.4% participants with BMI-defined obesity, 20.8% with WC-defined obesity, and 73.6% without obesity defined by either BMI or WC. Obesity was classified as non-obese (misclassified obesity) in 11.7% of participants if BMI or WC alone was used to define obesity. BMI-defined obesity (HR 2.69, 95%CI 2.48–2.92), WC-defined obesity (HR 2.28, 95%CI 2.10–2.48), and obesity defined by either BMI or WC (HR 2.53, 95%CI 2.33–2.74) were associated with an increased risk of TKA. Compared with those without obesity, participants with misclassified obesity had an increased risk of TKA (HR 2.06, 95%CI 1.85–2.30). 22.7% of TKA in the

**Funding:** The recruitment of the Melbourne Collaborative Cohort Study was funded by VicHealth and Cancer Council Victoria. This study was funded by a program grants from the National Health Medical Research Council (NHMRC; 209057, 251533, 396414, 623208), and was further supported by infrastructure provided by Cancer Council Victoria. SMH is the recipient of National Health and Medical Research Council (NHMRC) Early Career Fellowship (APP1142198), YW and AEW are the recipients of NHMRC Translating Research Into Practice (APP1168185 and APP1150102, respectively). The funders had no role in study design, data collection and analysis, decision to publish, or preparation of the manuscript.

**Competing interests:** The authors have declared that no competing interests exist.

community can be attributable to BMI-defined obesity, and a further 3.3% of TKA can be identified if WC was also used to define obesity.

## Conclusions

Both BMI and WC should be used to identify obese individuals who are at risk of TKA for osteoarthritis and should be targeted for prevention and treatment.

## Introduction

Obesity is an important modifiable risk factor for knee osteoarthritis (OA) [1]. Using an accurate and simple measure of obesity to identify individuals who are at higher risk of knee OA is important if we are to prevent the disease. Body mass index (BMI) is primarily used as a simple screening tool for obesity at the population level [2], but it has the limitation of not taking into account body fat distribution [3]. Waist circumference (WC) estimates central obesity and has been shown to be a better predictor for cardio-metabolic morbidity and premature mortality than BMI [3], particularly for people with lower BMI [4] and for women [5]. Discordance between BMI and WC in classifying individuals as obese has been demonstrated in a number of studies. For example, an Australian study reported that approximately 40% of individuals having WC obesity (WC $\geq$102 cm for men and $\geq$88 cm for women) were not obese with respect to BMI (BMI $\geq$30 kg/m$^2$) [6]. This discordance was greater in Chinese adults with 75.7% of those who were obese with respect to WC not defined as obese based on BMI [7].

Previous studies have examined BMI or WC obesity separately as a risk factor for knee OA [1, 3, 8] with no study taking into account the known discordance between BMI and WC in classifying obesity. This is likely to have misclassified some people who are obese defined by BMI but not WC, and vice versa, who may be at risk of knee OA. Examining both BMI and WC has the potential to better identify those at increased risk of knee OA and intervene accordingly. Thus we aimed to examine the risk of severe knee OA assessed by total knee arthroplasty (TKA) due to OA associated with obesity defined by BMI or WC and whether there is discordance between these measures in assessing this risk.

## Materials and methods

### Study population with inclusion and exclusion criteria

The Melbourne Collaborative Cohort Study (MCCS) is a well-established cohort study that recruited 41,514 participants (17,045 men, 99.3% aged 40–69 years) during 1990–1994 [9]. All subjects gave their informed written consent for inclusion before they participated in the study. The study was conducted in accordance with the Declaration of Helsinki, and the protocol was approved by the Cancer Council Victoria's Human Research Ethics Committee (IEC No. 9001) [9]. For the current study, 4693 (11.3%) were excluded because they had either: died or left Australia or reported having an arthroplasty prior to 1 January 2001; or their first recorded procedure was a total hip arthroplasty to a revision surgery [9]. We also excluded participants with missing data on BMI or WC (n = 37), leaving 36,784 participants available for analysis.

### Data collection on total knee arthroplasty for OA

The Australian Orthopaedic Association National Joint Replacement Registry (AOA NJRR) collects information on prostheses, patient demographics, type and reason for arthroplasty, with an almost complete data relating to arthroplasty (>99%) in Australia [10]. Linking the

MCCS records to the AOA NJRR identified those who had a primary TKA performed between 1 January 2001 and 31 December 2018. Knee OA was defined as the first primary TKA with a contemporaneous diagnosis of OA, as recorded in the AOA NJRR. If one person had multiple arthroplasties, the first recorded procedure was considered the event. The linkage study was approved by the Human Research Ethics Committee of Cancer Council Victoria (HREC 0601) and Monash University (2006000608).

### Data collection on demography and anthropometry

At baseline, demographic and lifestyle data, including date of birth, sex, and country of birth, were collected using pre-piloted standard questionnaires [11]. Smoking was assessed by asking participants if they smoked, if so whether they have smoked at least seven cigarettes a week, and were classified as non-smoker or current/ex-smoker [12, 13]. Physical activity was assessed by three separate questions obtained from the Risk Factor Prevalence Study conducted by the National Heart Foundation and Australian Institute of Health regarding frequency of non-occupational vigorous and moderate physical activity, and walking [12, 14]. Participants were categorised as whether or not participating in vigorous activity in line with values published in the Compendium of Physical Activities [15]. WC, height and weight were measured using standard procedures [9, 11]. Obesity was defined by BMI $\geq 30$ kg/m$^2$ or WC $\geq 102$ cm for men and $\geq 88$ cm for women [16]. Individuals with obesity based on either high BMI or high WC were identified. Obesity status was classified into three categories based on a combination of BMI and WC obesity: 'no obesity' (not having obesity based on either BMI or WC); 'obesity based on both BMI and WC'; 'misclassified obesity' if an individual is classified as non-obese when BMI or WC alone was used to define obesity.

### Statistical analysis

Cox proportional hazard regression models were used to estimate the hazard ratio (HR) and 95% confidence interval (CI) for the incidence of TKA due to OA associated with different definitions of obesity and obesity status, with age as the time scale. Follow-up for TKA (calculation of person-time) began 1 January 2001 and ended at the date of first TKA for OA or date of censoring. Participants were censored at either the date of first TKA indications other than OA, the date of death, or end of follow-up, whichever came first. To test whether the association between obesity and TKA for OA was modified by sex, interactions were fitted and tested using the likelihood ratio test. Since there was no interaction ($>0.20$), all analyses were performed for total population. However, as women have a higher prevalence of obesity and knee OA compared with men, sex stratified analyses were also performed. Population attributable fraction (PAF) was calculated to determine the proportion of knee arthroplasties in the population that could be attributable to obesity: obesity based on BMI, obesity based on WC, and obesity based on either BMI or WC, using the "punafcc" command in Stata, which implements the method recommended by Greenland and Drescher [17]. The formula for PAF used is $\sum pd_i$ [$(HR_i - 1)/HR_i$], where $pd_i$ is the proportion of TKA for OA observed in the i$^{th}$ obesity category and $HR_i$ is the hazard ratio (HR) associated with that category. The STATA formula for 'punafcc' is presented as an appendix (S1 Appendix). All analyses were adjusted for sex, smoking status, physical activity and country of birth. Confounders were selected based on the Directed acyclic graph (DAG) diagram [18] (S1 Fig) Tests based on Cox regression methods showed no evidence that proportional hazard assumptions were violated for any analysis. All statistical analyses were performed using Stata 15.0 (StataCorp LP., College Station, TX, USA).

## Results

Of the whole population (n = 36,784), 73.6% (n = 27,056) had no obesity, 26.4% (n = 9,728) had obesity based on either BMI or WC. When BMI was used to define obesity, 20.4% of the participants were classified as obese. If WC was used to define obesity, 20.8% were classified as obese. Obesity was misclassified in 11.7% (n = 4,290) of the participants if BMI or WC alone was used to define obesity (Fig 1).

Table 1 shows the general characteristics of the study participants. Over an average of 15.4 ±4.8 years' follow-up, 2,683 participants underwent TKA for OA. Those who had a TKA were older and more likely to be born in Australia/UK and had a higher BMI and WC than those without TKA. The prevalence of obesity defined by any definition (either BMI or WC, both BMI and WC) was higher among those who had a TKA.

Table 2 shows the relationship of different definitions of obesity and obesity status with the incidence of TKA for OA. After full adjustment, participants with obesity defined by BMI (HR 2.69, 95% CI 2.48, 2.92), WC (HR 2.28, 95% CI 2.10, 2.48), or defined by either BMI or WC (HR 2.53, 95% CI 2.33, 2.74) had an increased risk of TKA for OA compared with those who were not classified as obese using each of these definitions. Participants having obesity defined by both BMI and WC (HR 2.93, 95% CI 2.68, 3.21), and those with misclassified obesity, with only one of BMI or WC meeting criteria for obesity (HR 2.06, 95% CI 1.85, 2.30) had an increased risk of TKA for OA compared with those without either BMI or WC obesity. The relationship between different definitions of obesity and incidence of TKA for OA were similar in the sex-stratified analysis (S1 Table).

Table 3 shows the PAF of TKA in relation to different definitions of obesity, i.e. the estimated fraction of TKA that could be attributable to obesity. The PAF for obesity defined by either BMI or WC on TKA was estimated to be 26.0% (95% CI 23.5%, 28.5%) after adjustment for potential confounders. The PAFs for obesity based on BMI and obesity based on WC were 22.7% (95% CI 20.5%, 25.0) and 18.9% (95% CI 16.5%, 21.1%), respectively. In sex-stratified analysis the PAFs for obesity on TKA were higher in women than men (S2 Table).

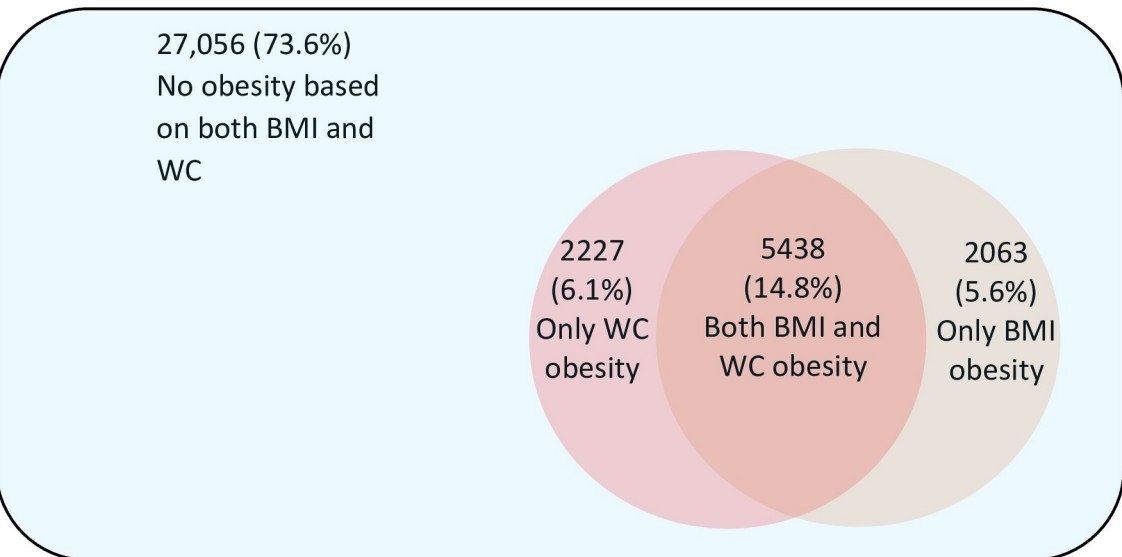

**Fig 1. Obesity defined by body mass index and/or waist circumference and discordance between these measures.**

**Table 1. Demographic characteristics and obesity status of study participants at baseline (1990–94).**

| | Total population (n = 36,784) | No knee arthroplasty (n = 34,101) | Knee arthroplasty (n = 2,683) | P value No knee arthroplasty vs Knee arthroplasty |
|---|---|---|---|---|
| Age, years | 62.6 (8.9) | 62.5 (8.9) | 63.7 (7.8) | <0.001 |
| (Range) | (35.3–83.0) | (35.3–83.0) | (45.8–79.9) | |
| Gender | | | | <0.001 |
| Male, n (%) | 14,888 (40.5) | 13,976 (41.0) | 912 (34.0) | |
| Female, n (%) | 21,896 (59.5) | 20,125 (59.0) | 1,771 (66.0) | |
| Country of birth, n (%) | | | | <0.001 |
| Australia and UK | 27,705 (75.3) | 25,493 (74.8) | 2,212 (82.4) | |
| Italy and Greek | 9,079 (24.7) | 8,608 (25.2) | 471 (17.6) | |
| Moderate and high level of physical activity, n (%) | 21,168 (57.6) | 19,569 (57.4) | 1,599 (59.6) | 0.03 |
| Current/ex-smoker, n (%) | 15,402 (41.9) | 14,394 (42.2) | 1,008 (37.6) | <0.001 |
| BMI, kg/m$^2$, mean (SD) | 26.9 (4.4) | 26.7 (4.3) | 29.1 (4.8) | <0.001 |
| (Range) | (14.0–57.8) | (14.0–57.8) | (18.0–49.7) | |
| Waist circumference, cm, mean (SD) | 85.2 (12.9) | 85.0 (12.8) | 88.8 (12.7) | <0.001 |
| (Range) | (47.0–166.5) | (47.0–166.5) | (55.0–135.5) | |
| Obesity based on BMI, n (%) | 7,501 (20.4) | 6,529 (19.4) | 973 (36.3) | <0.001 |
| Obesity based on WC, n (%) | 7,665 (20.8) | 6,745 (19.8) | 920 (34.3) | <0.001 |
| Obesity based on EITHER BMI or WC, n (%) | 9,728 (26.4) | 8,564 (25.1) | 1,164 (43.4) | <0.001 |
| Obesity status | | | | <0.001 |
| No obesity based on both BMI and WC | 27,056 (73.6) | 25,537 (74.9) | 1,519 (56.6) | |
| Obesity based on BOTH BMI and WC | 5,438 (14.8) | 4,709 (13.8) | 729 (27.2) | |
| Misclassified as non-obese if obesity was defined by either BMI or WC alone | 4,290 (11.7) | 3,855 (11.3) | 435 (16.2) | |

UK, United Kingdom; BMI, body mass index; WC, waist circumference; Obesity based on BMI: defined as BMI $\geq$30 kg/m$^2$; Obesity based on WC: defined as WC $\geq$88cm for men & $\geq$102cm for women.

## Discussion

We found obesity was misclassified in 11.7% of participants if BMI or WC alone was used to define obesity. All participants with obesity, regardless of definition used (BMI only, WC only, both BMI and WC, or either BMI or WC) had an increased risk of TKA for OA compared

**Table 2. Relationship of different definitions of obesity, and obesity status with incidence of total knee arthroplasty for osteoarthritis.**

| | Model 1 Hazard ratio (95% CI) | Model 2 Hazard ratio (95% CI) |
|---|---|---|
| Obesity based on BMI, yes/no | 2.32 (2.14, 2.51) | 2.69 (2.48, 2.92) |
| Obesity based on WC, yes/no | 2.05 (1.89, 2.23) | 2.28 (2.10, 2.48) |
| Obesity based on EITHER BMI or WC, yes/no | 2.21 (2.05, 2.39) | 2.53 (2.33, 2.74) |
| Obesity status | | |
| No obesity either BMI obesity or WC obesity | 1.00 | 1.00 |
| Obesity is not identified if one of BMI or WC is used | 1.83 (1.65, 2.04) | 2.06 (1.85, 2.30) |
| Obesity based on BOTH BMI and WC | 2.51 (2.30, 2.74) | 2.93 (2.68, 3.21) |

CI, confidence interval; BMI, body mass index; WC, waist circumference; Model 1. adjusted for age and sex, Model 2: adjusted for age, sex, smoking status, physical activity and country of birth.

**Table 3. Estimated population attributable fraction (PAF, %) of total knee arthroplasty in relation to different definitions of obesity.**

|  | Model 1 PAF (95% CI) | Model 2 PAF (95% CI) |
|---|---|---|
| **Obesity based on BMI** | 20.7 (18.4, 23.0) | 22.7 (20.5, 25.0) |
| **Obesity based on WC** | 17.2 (14.9, 20.0) | 18.9 (16.5, 21.1) |
| **Obesity based on either BMI and WC** | 23.7 (21.0, 26.2) | 26.0 (23.5, 28.5) |

PAF, population attributable fraction; CI, confidence interval; BMI, body mass index; WC, waist circumference; Model 1. adjusted for age and sex, Model 2: adjusted for age, sex, smoking status, physical activity and country of birth.

with those with no obesity. Almost 26% of TKAs in the population can be attributable to obesity defined by either BMI or WC. Whereas, if as is currently done, we defined obesity based on only BMI or WC, 3.3% and 7.1% of TKAs that can be attributable to obesity were missed, respectively. The results remained similar when we adjusted for age and sex (model 1) and further adjusted for smoking status, vigorous physical activity and country of birth (model 2). These relationships held true for both men and women, particularly for women.

We found that obesity based on either BMI or WC was associated with an increased risk of TKA for OA. Compared with non-obese participants, individuals with BMI obesity, those with WC obesity, and those with either BMI or WC obesity had approximately double the risk of TKA. However, the risk of TKA was somewhat highest in participants with obesity defined by both BMI and WC. Recently, for the identification of individuals at risk of knee OA, BMI has been suggested as a sufficient measure of obesity [19]. Although our data broadly support this, we found that 22.7% of TKAs in the community can be attributable to obesity defined by BMI, but a further 3.3% of this could be identified if obesity was also defined by WC, even in the absence of BMI-defined obesity. As we and others have shown that WC is also a risk factor for knee arthroplasty for OA [1, 3, 8], this represents a significant missed opportunity to identify and target those at risk of knee arthroplasty for OA. This additional 3.3% is significant given that 22.7% of TKA for OA are attributable to BMI obesity. Thus our study supports using both BMI and WC to identify individuals with obesity who are at risk of knee OA for disease prevention and management. Given the lack of effective therapies to prevent disease progression in knee OA, this is of particular importance as we grapple with improved methods of prevention.

The strengths of our study include its prospective design, large sample size, participants of varying age and country of birth, and the validation and completeness of arthroplasty data from the AOA NJRR [10]. Our results need to be considered within the study's limitations. In this study, we have used TKA as a proxy for severe symptomatic OA. Although, other factors such as access to health care, patient and clinician factors influence the decision for arthroplasty [20], the publicly-funded universal health system (Medicare) in Australia ensures that everyone has access to arthroplasty facilities. Our analyses have controlled for age, sex, smoking status, physical activity and country of birth. Using TKA as a surrogate measure of knee OA, we have shown consistent associations between established risk factors for knee OA and the risk of TKA for OA [3, 9, 21]. The similar procedure is used as a validated measure of defining knee OA in the Scandinavian countries as the data are available in reliable national registries [8]. Arthroplasty data were not available prior to 2001 and as a result, some misclassification of arthroplasty may have occurred. This is most likely to have been non-differential which might have attenuated the strength of the observed associations. The assumption for PAF calculation is that there is a causal relationship between the risk factor and outcome. PAF

can be calculated using data from observational studies as long as this assumption is considered [22]. A further assumption in PAF is that the distribution of the other confounders regardless of obesity remains stable. Our key confounders of age, sex, and country of birth are stable. Studies have shown that there is a relatively strong concordance of levels of obesity and physical activity status over 20 years [23, 24]. However, some misclassification may have occurred.

## Conclusions

Both BMI and WC should be used to define obesity in order to identify those at risk of knee OA, as both measures are associated with an increased risk of TKA for OA. Defining obesity using BMI as well as WC identifies a higher proportion of those with obesity related severe knee OA with the potential to improve approaches to the prevention of knee OA.

## Supporting information

**S1 Fig. The relationship of obesity and the confounders with total knee replacement for osteoarthritis.**
(TIF)

**S1 Table. Relationship of different definitions of obesity, and obesity status with incidence of total knee arthroplasty for osteoarthritis.**
(DOCX)

**S2 Table. Estimated population attributable fraction (PAF, %) of total knee arthroplasty in relation to different definitions of obesity.**
(DOCX)

**S1 Appendix. STATA formula for 'punafcc'.**
(DOCX)

## Acknowledgments

The Melbourne Collaborative Cohort Study was made possible by the contribution of many people, including the original investigators and the diligent team who recruited the participants and who continue working on follow up. We would like to express our gratitude to the many thousands of Melbourne residents who participated in the study. For the data linkage, we would especially like to thank the Registry coordinator Ann Tomkins and statistician Lisa Miller from the Australian Orthopaedic Association National Joint Replacement Registry, and Ms Georgina Marr from Cancer Council Victoria.

## Author Contributions

**Conceptualization:** Yuanyuan Wang, Flavia M. Cicuttini, Sultana Monira Hussain.

**Data curation:** Graham G. Giles, Stephen Graves.

**Formal analysis:** Sultana Monira Hussain.

**Methodology:** Yuan Z. Lim, Yuanyuan Wang, Flavia M. Cicuttini, Stephen Graves, Anita E. Wluka, Sultana Monira Hussain.

**Supervision:** Sultana Monira Hussain.

**Validation:** Graham G. Giles, Stephen Graves.

**Writing – original draft:** Yuan Z. Lim, Yuanyuan Wang, Sultana Monira Hussain.

**Writing – review & editing:** Yuan Z. Lim, Yuanyuan Wang, Flavia M. Cicuttini, Graham G. Giles, Stephen Graves, Anita E. Wluka, Sultana Monira Hussain.

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
