## [Decision Letter · Decision Letter 0]

9 Sep 2020

PONE-D-20-22684

Obesity defined by body mass index and waist circumference and risk of total knee arthroplasty for osteoarthritis: a prospective cohort study

PLOS ONE

Dear Dr. Hussain,

Thank you for submitting your manuscript to PLOS ONE. After careful consideration, we feel that it has merit but does not fully meet PLOS ONE’s publication criteria as it currently stands. Therefore, we invite you to submit a revised version of the manuscript that addresses the points raised during the review process.

We look forward to receiving your revised manuscript.

Kind regards,

Osama Farouk

Academic Editor

PLOS ONE

Journal Requirements:

3.Thank you for stating the following financial disclosure:

 [The funders had no role in study design, data collection and analysis, decision to publish, or preparation of the manuscript.].

Reviewers' comments:

Reviewer's Responses to Questions

**Comments to the Author**

1. Is the manuscript technically sound, and do the data support the conclusions?

Reviewer #1: Yes

Reviewer #2: Yes

Reviewer #3: Yes

Reviewer #4: Yes

2. Has the statistical analysis been performed appropriately and rigorously? 

Reviewer #1: Yes

Reviewer #2: Yes

Reviewer #3: Yes

Reviewer #4: Yes

3. Have the authors made all data underlying the findings in their manuscript fully available?

Reviewer #1: Yes

Reviewer #2: No

Reviewer #3: Yes

Reviewer #4: No

4. Is the manuscript presented in an intelligible fashion and written in standard English?

Reviewer #1: Yes

Reviewer #2: Yes

Reviewer #3: Yes

Reviewer #4: Yes

5. Review Comments to the Author

Reviewer #1: Congratulations for the great work. Having access to this information bank of your country is a privilege that should be used as you did. Obesity has to be combated by all health professionals and cannot go unnoticed. Making this risk clear is really our role.

Reviewer #2: Using a longitudinal study, the authors demonstrated that obesity is a risk factor for total knee arthroplasty for osteoarthritis. They also estimated PAR for obesity based on its different definition. Here are my concerns.

Method

• The authors stated that they employed a standard questionnaire and cited the reference number 9. Please provide the reliability and validity indices of the study questionnaire.

• The classification of obesity based on WC and BMI is not clear. Please using a 2 by 2 table, clarify the status of the participants based on these 2 index.

• Please provide the confounding criteria utilized for study confounder selection?

• Please add the utilized PARF formula.

Results

• Table 1: please add the utilized test for comparison as well as their corresponding Pvalue.

Study limitation:

• Please note that the interpretation of PAF need to consider its well-known assumptions i.e. 1) the estimated ORs should be causal (which is not supported in observational studies), 2) the distribution of the other confounders regardless of obesity must be stable.

Reviewer #3: The paper is well writen and represents useful information for the clinician. BMI is frequently used for obesity screening. However, waist circumference seems more adequate in the evaluation of fat distribution. This paper indicates that, regarding knee OA, both measures indicate patients with higher risk of TKA for osteoarthritis. So, both can be used in clinical assessment.

Reviewer #4: General comments:

Th idea about “ Obesity defined by body mass index and waist circumference and risk of total knee arthroplasty for osteoarthritis” is interesting and original. Obesity is an important modifiable and on the other hand knee osteoarthritis as a common and significant orthopedic problem with subsequent knee arthroplasty. As a whole this manuscript is almost well written. However, I have some comments below:

Abstract:

The statement objective is not written well, it’s better to be written as that was at the end of the introduction.

Methods:

- The study design is a good and strength point and support the results of the study in addition to the large sample size.

- The subtitles of the methods is not suitable for this section, “Incidence of total knee arthroplasty for OA” and “Demographic data, anthropometric measurements and classification of obesity status” could be written in results section and not as a methodology subtitle. The subtitles in this section are study population with inclusion and exclusion criteria, Data collection as sampling, data collection tools,….etc

- Some details about the assessment of physical activity, questionnaires used are needed

- The absence of the flow chart of this cohort study is a great missing and should be included.

- Inclusion of other factors as risk factors as secondary outcomes as: sex, age, physical activity and residence could be beneficial in this large study with large data.

Results:

- The titles of the tables and figures are short and are needed to be written in complete informative way especially of table (1).

- In table (1):

• text comment there is a discrepancy between the total number of of participants with TKA and the number in table (1), please correct.

• It’s better to include female number with male number

• Clarify the numbers related to the BMI: AS if they are about mean+ SD, or what are they refer to? The same is in waist circumference.

• Add the range (minimum – maximum) in age, BMI and waist circumference.

Discussion

- After inclusion of other factors ( sex, age, physical activity and residence) could be included in the discussion section.

6. PLOS authors have the option to publish the peer review history of their article (what does this mean?). If published, this will include your full peer review and any attached files.

Reviewer #1: **Yes: **Joao Paulo Fernandes Guerreiro (Guerreiro, JPF)

Reviewer #2: No

Reviewer #3: No

Reviewer #4: **Yes: **Dalia G Mahran

---

## [Author Response · Author response to Decision Letter 0]

21 Oct 2020

PONE-D-20-22684

Obesity defined by body mass index and waist circumference and risk of total knee arthroplasty for osteoarthritis: a prospective cohort study

Review Comments to the Author

Reviewer #1: Congratulations for the great work. Having access to this information bank of your country is a privilege that should be used as you did. Obesity has to be combated by all health professionals and cannot go unnoticed. Making this risk clear is really our role.

Response: We would like to thank the reviewer for these encouraging comments.

Reviewer #2: Using a longitudinal study, the authors demonstrated that obesity is a risk factor for total knee arthroplasty for osteoarthritis. They also estimated PAR for obesity based on its different definition. Here are my concerns.

Response: We would like to respond in line with the reviewer’s comments.

Method

• The authors stated that they employed a standard questionnaire and cited the reference number 9. Please provide the reliability and validity indices of the study questionnaire.

Response: Data on date of birth, sex, country of birth, and education were collected based on standard questionnaires that were piloted prior to data collection(1). Smoking was assessed by asking participants if they currently smoke. Participants who reported currently smoking at least seven cigarettes weekly were categorised as current smokers. Those not currently smoking but who had smoked at least seven cigarettes weekly for at least a year were categorised as ex‐smokers. Others were classified as never smokers(2, 3). Physical activity was assessed by three separate questions obtained from the Risk Factor Prevalence Study conducted by the National Heart Foundation and Australian Institute of Health regarding frequency of non-occupational vigorous and moderate physical activity, and walking(2, 4). Participants were categorised as whether or not participating in vigorous physical activity in line with values published in the Compendium of Physical Activities(5). All these suggest the reliability and validity of the study questionnaire.

Author action: We have now clarified the issue raised by the reviewer and amended the text in the manuscript. (page 5, paragraph 1)

At baseline, demographic and lifestyle data, including date of birth, sex, and country of birth, were collected using pre-piloted standard questionnaires (11). Smoking was assessed by asking participants if they smoked, if so whether they have smoked at least seven cigarettes a week, and were classified as non-smoker or current/ex-smoker(12, 13). Physical activity was assessed by three separate questions obtained from the Risk Factor Prevalence Study conducted by the National Heart Foundation and Australian Institute of Health regarding frequency of non-occupational vigorous and moderate physical activity, and walking(12, 14). Participants were categorised as whether or not participating in vigorous activity in line with values published in the Compendium of Physical Activities(15). 

• The classification of obesity based on WC and BMI is not clear. Please using a 2 by 2 table, clarify the status of the participants based on these 2 index.

Response: As we have indicated in our manuscript, we firstly defined obesity based on BMI (BMI ≥30 kg/m2) and WC (WC ≥102 cm for men and ≥88 cm for women) separately. Based on these two variables obesity status was classified as ‘no obesity’ (not having obesity based on either BMI or WC); ‘obesity based on both BMI and WC’; and ‘misclassified obesity’ (obesity based on either BMI or WC, but not both). As the reviewer requested, we present the 2 by 2 table below. 

 Obesity based on WC 

No Obesity based on WC 

Yes Total

Obesity based on BMI 

No 27056 2227 29283

Obesity based on BMI 

Yes 2063 5438 7501

Total 29119 7665 36784

There were 27,056 participants who did not have obesity defined by either BMI or WC, and 5438 participants were defined as obesity based on both BMI and WC. 

Based on BMI only, 7501 participants had obesity. However, 2227 participants who were classified as non-obese based on BMI were obese based on WC.

Based on WC only, 7665 participants had obesity. However, 2063 participants who were classified as non-obese based on WC were obese based on BMI.

So 4290 participants were misclassified if only BMI or WC was used to define obesity. 

Author action: We have presented these numbers and categories in our manuscript as Figure 1.

 

Fig 1: Obesity defined by body mass index and/or waist circumference and discordance between these measures

• Please provide the confounding criteria utilized for study confounder selection?

Response: The figure below explains the relationship between obesity and total knee replacement for osteoarthritis with all variables used to build the models. We have selected the model based on the DAG diagram(6).

Author action: We have added the following text (page 6, paragraph 1) and have included the figure as Supplementary Figure in the manuscript.

Confounders were selected based on the Directed acyclic graph (DAG) diagram(6) (S1 Fig)

 

S1 Fig: The relationship of obesity and the confounders with total knee replacement for osteoarthritis

For our model building, we first considered the model which included all the predictors that had a p-value of less than 0.2 – 0.25 in the univariate analyses which in this particular analysis meant that we included every predictor in our model (https://stats.idre.ucla.edu/stata/seminars/stata-survival/#building).

Table: the unadjusted association of the following variables with knee replacement 

Explanatory variable HR (95% CI) P value

Sex 1.26 (1.16, 1.37) <0.001

Smoking status 0.89 (0.82, 0.96) 0.003

Vigorous physical activity 1.09 (1.01, 1.18) 0.02

Country of birth 0.60 (0.54, 0.67) <0.001

Education 1.01 (0.94, 1.10) 0.75

All the variables except for education fulfilled these criteria for inclusion in the final regression model. We repeated our analysis without adjustment for education, the results were very similar. 

Author action: We have removed education from the list of confounders. Now all our analyses are adjusted for age, sex, smoking status, vigorous physical activity and country of birth. 

• Please add the utilized PARF formula.

Response: We used the “punafcc” command from STATA that estimates a marginal mean between-scenario risk or hazard ratio for survival data. The formula for PAF is: 

∑pdi[(HRi − 1)/HRi], where pdi is the proportion of total knee replacement for OA observed in the ith obesity category and HRi is the hazard ratio (HR) associated with that category.

In STATA the formula looks like the following 

This implies that is the population mean risk ratio (or hazard ratio) between scenario i and the real world for the “subsubpopulation” of cases (or failures) of the subpopulation specified by the subpop() option and that (4) is a corresponding sample mean risk ratio (or hazard ratio) for the “subsubsample” of cases (or failures) of the subsample specified by the subpop() option. A mean between-scenario ratio is a subtly different quantity from a ratio between scenario means; however, both of these quantities are known as population unattributable fractions and can be subtracted from 1 to give population attributable fractions.

Author action: We have added the following line in the statistical analysis section (page 6, paragraph 1). We have also presented the STATA formula as an appendix.

The formula for PAF used is: 

∑pdi[(HRi − 1)/HRi], where pdi is the proportion of total knee replacement for OA observed in the ith obesity category and HRi is the hazard ratio (HR) associated with that category. The STATA formula for ‘punafcc’ is presented as an appendix (S1 Appendix). 

Results

• Table 1: please add the utilized test for comparison as well as their corresponding Pvalue.

Response: According to current STROBE guideline for reporting cohort studies, “P values for the table that describes the cohort profile are not essential”(7, 8). In fact in the study published in the PLoS Medicine and the other studies regarding reporting STROBE checklist published in the Lancet, Epidemiology, Annals of Internal Medicine, and the BMJ, a P value was not presented for the descriptive tables they have provided as an example. However, as suggested by the reviewer we have included p values in table 1.

Table 1. Baseline characteristics of study participants 

 Total population

(n=36,784) No knee arthroplasty (n=34,101) Knee arthroplasty

(n=2,683) P value

No knee arthroplasty vs Knee arthroplasty

Age, years 62.6 (8.9) 62.5 (8.9) 63.7 (7.8) <0.001

Male, n (%) 14,888 (40.5) 13,976 (41.0) 912 (34.0) <0.001

Country of birth, n (%) <0.001

 Australia and UK 27,705 (75.3) 25,493 (74.8) 2,212 (82.4) 

 Italy and Greek 9,079 (24.7) 8,608 (25.2) 471 (17.6) 

Secondary education, degree/diploma, n (%) 15,606 (42.7) 14,503 (42.8) 1,103 (41.3) 0.12

Moderate and high level of physical activity, n (%) 21,168 (57.6) 19,569 (57.4) 1,599 (59.6) 0.03

Current/ex-smoker, n (%) 15,402 (41.9) 14,394 (42.2) 1,008 (37.6) <0.001

BMI, kg/m2 26.9 (4.4) 26.7 (4.3) 29.1 (4.8) <0.001

Waist circumference, cm 85.2 (12.9) 85.0 (12.8) 88.8 (12.7) <0.001

Obesity based on BMI, n (%) 7,501 (20.4) 6,529 (19.4) 973 (36.3) <0.001

Obesity based on WC, n (%) 7,665 (20.8) 6,745 (19.8) 920 (34.3) <0.001

Obesity based on EITHER BMI or WC, n (%) 9,728 (26.4) 8,564 (25.1) 1,164 (43.4) <0.001

Obesity status <0.001

 No obesity based on both BMI and WC 27,056 (73.6) 25,537 (74.9) 1,519 (56.6) 

 Obesity based on BOTH BMI and WC 5,438 (14.8) 4,709 (13.8) 729 (27.2) 

 Misclassified as non-obese if obesity was defined by either BMI or WC alone 4,290 (11.7) 3,855 (11.3) 435 (16.2) 

UK, United Kingdom; BMI, body mass index; WC, waist circumference; Obesity based on BMI: defined as BMI ≥30 kg/m2; Obesity based on WC: defined as WC ≥88cm for men & ≥102cm for women

Study limitation:

• Please note that the interpretation of PAF need to consider its well-known assumptions i.e. 1) the estimated ORs should be causal (which is not supported in observational studies), 2) the distribution of the other confounders regardless of obesity must be stable.

Response: We agree with the reviewer that the assumption for PAF is that there is a causal relationship between the risk factors and the outcome. This is the assumption inherent whenever a PAF is calculated(9). PAF is most commonly calculated from data derived from observational studies despite this limitation. In our study, age, sex, country of birth, education all remained stable. Smoking status might change over time. However, this is a group of community-based population, and thus without any specific indication changes in smoking habit is low. Previous studies have suggested a relatively strong concordance of weight(10) and physical activity(11) over 10-20 years especially in a stable older population. 

Author action: We have included the following statements in the limitations section (page 12, paragraph 1)

The assumption for PAF calculation is that there is a causal relationship between the risk factor and outcome. PAF can be calculated using data from observational studies as long as this assumption is considered(22). A further assumption in PAF is that the distribution of the other confounders regardless of obesity remains stable. Our key confounders of age, sex, and country of birth are stable. Studies have shown that there is a relatively strong concordance of levels of obesity and physical activity status over 20 years(23, 24). However, some misclassification may have occurred.

Reviewer #3: The paper is well writen and represents useful information for the clinician. BMI is frequently used for obesity screening. However, waist circumference seems more adequate in the evaluation of fat distribution. This paper indicates that, regarding knee OA, both measures indicate patients with higher risk of TKA for osteoarthritis. So, both can be used in clinical assessment.

Response: We would like to thank the reviewer for these comments

Reviewer #4: General comments:

The idea about “ Obesity defined by body mass index and waist circumference and risk of total knee arthroplasty for osteoarthritis” is interesting and original. Obesity is an important modifiable and on the other hand knee osteoarthritis as a common and significant orthopedic problem with subsequent knee arthroplasty. As a whole this manuscript is almost well written. However, I have some comments below:

Response: We would like to thank the reviewer for these encouraging comments.

Abstract:

The statement objective is not written well, it’s better to be written as that was at the end of the introduction.

Author action: We have amended the aims in the abstract (page 2, paragraph 1). 

To examine the risk of total knee arthroplasty (TKA) due to osteoarthritis associated with obesity defined by body mass index (BMI) or waist circumference (WC) and whether there is discordance between these measures in assessing this risk. 

Methods:

- The study design is a good and strength point and support the results of the study in addition to the large sample size.

Response: Thank you for the encouraging words.

- The subtitles of the methods is not suitable for this section, “Incidence of total knee arthroplasty for OA” and “Demographic data, anthropometric measurements and classification of obesity status” could be written in results section and not as a methodology subtitle. The subtitles in this section are study population with inclusion and exclusion criteria, Data collection as sampling, data collection tools,….etc

Response: We have changed the subtitles as suggested by the reviewer. 

Author action: We have changed the subtitles of the method section as suggested by the reviewer. Our current subtitles for the methods section are-

Study population with inclusion and exclusion criteria 

Data collection on total knee arthroplasty for OA 

Data collection on demography and anthropometry

- Some details about the assessment of physical activity, questionnaires used are needed

Response: Physical activity was assessed by three separate questions obtained from the Risk Factor Prevalence Study conducted by the National Heart Foundation and Australian Institute of Health regarding frequency of non-occupational vigorous and moderate physical activity, and walking(2, 4). Participants were categorised as whether or not participating in vigorous physical activity in line with values published in the Compendium of Physical Activities(5). 

Author action: We have now clarified the issue raised by the reviewer and have amended the text in the manuscript. (page 5, paragraph 1)

Physical activity was assessed by three separate questions obtained from the Risk Factor Prevalence Study conducted by the National Heart Foundation and Australian Institute of Health regarding frequency of non-occupational vigorous and moderate physical activity, and walking(12, 14). Participants were categorised as whether or not participating in vigorous activity in line with values published in the Compendium of Physical Activities(15). 

- The absence of the flow chart of this cohort study is a great missing and should be included.

Response: As we have included ~90% of the original participants in the statistical analysis and we have few missing data, we did not include a flow chart. As the reviewer has mentioned, the flow chart is given below.

- Inclusion of other factors as risk factors as secondary outcomes as: sex, age, physical activity and residence could be beneficial in this large study with large data.

Response: We have included all these variables as confounders in the statistical analysis. Our aim was to examine the risk of severe knee OA assessed by total knee arthroplasty (TKA) due to OA associated with obesity defined by body mass index (BMI) or waist circumference (WC) and whether there is discordance between these measures in assessing this risk. We have examined the associations between exposure (obesity) and outcome (knee arthroplasty for OA) based on two regression models. Model 1 adjusted for age and sex, and Model 2 adjusted for age, sex, smoking status, vigorous physical activity and country of birth.

Results:

- The titles of the tables and figures are short and are needed to be written in complete informative way especially of table (1).

Response: As suggested, we have elaborated the title of table one. (page 7)

Table 1. Demographic characteristics and obesity status of study participants at baseline (1990-94)

- In table (1):

• text comment there is a discrepancy between the total number of of participants with TKA and the number in table (1), please correct.

Response: Thank you for pointing this out. This was an unintentional mistake. We have corrected this in the text.

• It’s better to include female number with male number

Response: As suggested, we have added the number of females in the table

 Total population

(n=36,784) No knee arthroplasty (n=34,101) Knee arthroplasty

(n=2,683)

Age, years 62.6 (8.9) 62.5 (8.9) 63.7 (7.8)

Male, n (%) 14,888 (40.5) 13,976 (41.0) 912 (34.0)

Female, n (%) 21,896 (59.5) 20,125 (59.0) 1,771 (66.0)

• Clarify the numbers related to the BMI: AS if they are about mean+ SD, or what are they refer to? The same is in waist circumference.

Response: As suggested by the reviewer, we have clarified this in the table.

This reads 

BMI, kg/m2, mean (SD) 26.9 (4.4) 26.7 (4.3) 29.1 (4.8)

Waist circumference, cm, mean (SD) 85.2 (12.9) 85.0 (12.8) 88.8 (12.7)

• Add the range (minimum – maximum) in age, BMI and waist circumference.

Response: We checked the distribution of data on age, BMI and waist circumference. As these data were normally distributed, we presented mean and standard deviation. Several statistical publications suggested that, range is based on only two of the observations and may not be representative of the whole dataset, particularly if there are outliers which is very likely to be present in a large data set(12, 13). However, as the reviewer requested, we have presented these as an appendix.

Author action: The ranges of age, BMI and WC are presented in S2 Appendix.

S2 Appendix: The range (minimum – maximum) of age, body mass index and waist circumference

 Total population

(n=36,784) No knee arthroplasty (n=34,101) Knee arthroplasty

(n=2,683)

Age, years 35.3 - 83.0 35.3 - 83.0 45.8 - 79.9

Body mass index, kg/m2 14.0 - 57.8 14.0 - 57.8 18.0 - 49.7

Waist circumference, cm 47.0 - 166.5 47.0 - 166.5 55.0 - 135.5

Discussion

- After inclusion of other factors ( sex, age, physical activity and residence) could be included in the discussion section.

Response: As suggested by the reviewer, we have added the following lines in the discussion section.

Author action: (page 10, paragraph 2)

The results remained similar when we adjusted for age and sex (model 1) and further adjusted for smoking status, vigorous physical activity and country of birth (model 2).

1. Milne RL, Fletcher AS, MacInnis RJ, Hodge AM, Hopkins AH, Bassett JK, et al. Cohort Profile: The Melbourne Collaborative Cohort Study (Health 2020). Int J Epidemiol 2017;46:1757-i

2. Jayasekara H, English DR, Haydon A, Hodge AM, Lynch BM, Rosty C, et al. Associations of alcohol intake, smoking, physical activity and obesity with survival following colorectal cancer diagnosis by stage, anatomic site and tumor molecular subtype. International journal of cancer 2018;142:238-50

3. Siahpush M, English D, Powles J. The contribution of smoking to socioeconomic differentials in mortality: results from the Melbourne Collaborative Cohort Study, Australia. J Epidemiol Community Health 2006;60:1077-9

4. MacInnis RJ, English DR, Hopper JL, Haydon AM, Gertig DM, Giles GG. Body size and composition and colon cancer risk in men. Cancer Epidemiol Biomarkers Prev 2004;13:553-9

5. Ainsworth BE, Haskell WL, Leon AS, Jacobs DR, Jr., Montoye HJ, Sallis JF, et al. Compendium of physical activities: classification of energy costs of human physical activities. Medicine and science in sports and exercise 1993;25:71-80

6. Shrier I, Platt RW. Reducing bias through directed acyclic graphs. BMC Med Res Methodol 2008;8:70

7. Vandenbroucke JP, von Elm E, Altman DG, Gøtzsche PC, Mulrow CD, Pocock SJ, et al. Strengthening the Reporting of Observational Studies in Epidemiology (STROBE): explanation and elaboration. International journal of surgery (London, England) 2014;12:1500-24

8. Vandenbroucke JP, von Elm E, Altman DG, Gøtzsche PC, Mulrow CD, Pocock SJ, et al. Strengthening the Reporting of Observational Studies in Epidemiology (STROBE): explanation and elaboration. Annals of internal medicine 2007;147:W163-94

9. Mansournia MA, Altman DG. Population attributable fraction. BMJ (Clinical research ed) 2018;360:k757

10. Hruby A, Hu FB. The Epidemiology of Obesity: A Big Picture. PharmacoEconomics 2015;33:673-89

11. Guthold R, Stevens GA, Riley LM, Bull FC. Worldwide trends in insufficient physical activity from 2001 to 2016: a pooled analysis of 358 population-based surveys with 1·9 million participants. The Lancet Global health 2018;6:e1077-e86

12. Whitley E, Ball J. Statistics review 1: presenting and summarising data. Critical care (London, England) 2002;6:66-71

13. Soyemi K. Choosing the right statistical test. Pediatrics in review 2012;33:e38-44

---

## [Decision Letter · Decision Letter 1]

30 Nov 2020

PONE-D-20-22684R1

Obesity defined by body mass index and waist circumference and risk of total knee arthroplasty for osteoarthritis: a prospective cohort study

PLOS ONE

Dear Dr. Hussain,

Thank you for submitting your manuscript to PLOS ONE. After careful consideration, we feel that it has merit but does not fully meet PLOS ONE’s publication criteria as it currently stands. Therefore, we invite you to submit a revised version of the manuscript that addresses the points raised during the review process.

We look forward to receiving your revised manuscript.

Kind regards,

Osama Farouk

Academic Editor

PLOS ONE

Reviewers' comments:

Reviewer's Responses to Questions

**Comments to the Author**

1. If the authors have adequately addressed your comments raised in a previous round of review and you feel that this manuscript is now acceptable for publication, you may indicate that here to bypass the “Comments to the Author” section, enter your conflict of interest statement in the “Confidential to Editor” section, and submit your "Accept" recommendation.

Reviewer #4: All comments have been addressed

2. Is the manuscript technically sound, and do the data support the conclusions?

Reviewer #4: Yes

3. Has the statistical analysis been performed appropriately and rigorously? 

Reviewer #4: Yes

4. Have the authors made all data underlying the findings in their manuscript fully available?

Reviewer #4: (No Response)

5. Is the manuscript presented in an intelligible fashion and written in standard English?

Reviewer #4: Yes

6. Review Comments to the Author

Reviewer #4: Dear authors

Congratulations for your done valuable work. I have a short comment, it's better to add the range of age, BMI and waist circumference to table (1) as separate raws with each one under the related variable or between brackets under the related means and SD but not in a separate table. The normal distributed variables are presented as mean, SD and range (minimum and maximum), while the non parametric data are presented as median and interquartile range.

Dood luck

7. PLOS authors have the option to publish the peer review history of their article (what does this mean?). If published, this will include your full peer review and any attached files.

Reviewer #4: **Yes: **Dalia G Mahran

---

## [Author Response · Author response to Decision Letter 1]

1 Dec 2020

Response: Thank you for giving the opportunity to revise. We are submitting the revised version.

Response: 

• We have uploaded a rebuttal letter. The file is labelled as “response to reviewers”

• We have uploaded a marked copy of the revised manuscript labelled as 'Revised Manuscript with Track Changes'.

• We have also submitted a unmarked version of the manuscript labelled as ‘manuscript’

Response: Not applicable

Response: We want to report our financial disclosure as was reported in the cover letter when we submitted the manuscript for the first time. We have uploaded our figure 1 file in the PACE. PACE software reported “DOCX file is converted to a valid TIF file”. We are replacing the previous file with this current TIF file after downloading from PACE. Please let us know if we need to perform further modification. 

Reviewer #4: Dear authors

Congratulations for your done valuable work. I have a short comment, it's better to add the range of age, BMI and waist circumference to table (1) as separate raws with each one under the related variable or between brackets under the related means and SD but not in a separate table. The normal distributed variables are presented as mean, SD and range (minimum and maximum), while the non parametric data are presented as median and interquartile range.

Dood luck

Response: As the reviewer suggested, we have added the range of age, BMI and waist circumference to table (1) between brackets under the related means.

---

## [Decision Letter · Decision Letter 2]

21 Dec 2020

Obesity defined by body mass index and waist circumference and risk of total knee arthroplasty for osteoarthritis: a prospective cohort study

PONE-D-20-22684R2

Dear Dr. Hussain,

We’re pleased to inform you that your manuscript has been judged scientifically suitable for publication and will be formally accepted for publication once it meets all outstanding technical requirements.

Kind regards,

Osama Farouk

Academic Editor

PLOS ONE

Additional Editor Comments (optional):

Reviewers' comments:

Reviewer's Responses to Questions

**Comments to the Author**

1. If the authors have adequately addressed your comments raised in a previous round of review and you feel that this manuscript is now acceptable for publication, you may indicate that here to bypass the “Comments to the Author” section, enter your conflict of interest statement in the “Confidential to Editor” section, and submit your "Accept" recommendation.

Reviewer #4: All comments have been addressed

2. Is the manuscript technically sound, and do the data support the conclusions?

Reviewer #4: Yes

3. Has the statistical analysis been performed appropriately and rigorously? 

Reviewer #4: Yes

4. Have the authors made all data underlying the findings in their manuscript fully available?

Reviewer #4: (No Response)

5. Is the manuscript presented in an intelligible fashion and written in standard English?

Reviewer #4: Yes

6. Review Comments to the Author

Reviewer #4: (No Response)

7. PLOS authors have the option to publish the peer review history of their article (what does this mean?). If published, this will include your full peer review and any attached files.

Reviewer #4: No

---

## [Editor Report · Acceptance letter]

23 Dec 2020

PONE-D-20-22684R2 

Obesity defined by body mass index and waist circumference and risk of total knee arthroplasty for osteoarthritis: a prospective cohort study 

Dear Dr. Hussain:

I'm pleased to inform you that your manuscript has been deemed suitable for publication in PLOS ONE. Congratulations! Your manuscript is now with our production department. 

Kind regards, 

on behalf of

Dr. Osama Farouk 

Academic Editor

PLOS ONE